# Obscure to Observe: A Lesion-Aware MAE for Glaucoma Detection from Retinal Context

**Siddhant Bharadwaj**[1]                          siddhantb@iisc.ac.in
**Pratinav Seth**[2]                          seth.pratinav@gmail.com
**Chandra Sekhar Seelamantula**[1]                          css@iisc.ac.in
[1] *Dept. of Electrical Engineering, Indian Institute of Science Bengaluru, India*
[2] *AryaXAI Alignment Lab, Arya.ai, Mumbai, India.*

## Abstract

Self-supervised learning (SSL) offers a powerful paradigm for medical image representation learning, particularly in low-label regimes. However, standard pretext tasks often overlook domain-specific cues vital for diseases like glaucoma—a leading cause of irreversible blindness that manifests as subtle structural changes in the optic disc (OD) region. Understanding the broader retinal context is essential, yet traditional models tend to overfit to localized features, limiting generalizability. We propose a glaucoma-aware SSL framework using a Deconvolutional Masked Autoencoder (Deconv-MAE) with a ViT-B encoder, trained to reconstruct clean fundus images from inputs degraded by Gaussian noise and anatomically-aware OD masking. This lesion-focused corruption compels the model to learn robust, context-rich representations. Pretrained on EYEPACS and fine-tuned on ORIGA-light, our method outperforms both standard MAE and supervised baselines, highlighting the value of anatomically informed pretext tasks in retinal diagnostics.

**Keywords:** Retinal Fundus Images, Glaucoma Classification, Self-Supervised Learning

## 1. Introduction

Glaucoma is a leading cause of irreversible blindness (Kingman, 2004), making early detection critical. Retinal fundus imaging is the most accessible method for accurate diagnosis, as glaucoma is characterized by an enlarged optic cup due to increased pressure on the optic nerve. Various image processing and deep learning techniques have been developed for glaucoma detection (Ashtari-Majlan et al., 2024), but they typically rely on large annotated datasets—making the labor-intensive annotation process a major bottleneck. Self-Supervised Learning (SSL) (Chen et al., 2020; Grill et al., 2020; Caron et al., 2021) has emerged as a promising approach to to mitigate the dependence on labeled data. Among generative SSL methods, Masked Autoencoders (MAEs) (He et al., 2021)—which learn to reconstruct masked portions of images—have demonstrated success in various vision tasks (Zhang et al., 2022), their application to fundus imaging, particularly for glaucoma detection, remains limited. This limitation arises because many existing approaches do not fully exploit the unique anatomical and pathological features inherent in fundus images (Yang et al., 2024), such as the optic disc, which are critical for accurate glaucoma diagnosis.

To address this gap, we propose a domain-aware self-supervised learning framework tailored for glaucoma detection from fundus images. Our method explicitly masks the optic disc (OD)—the key region examined in clinical diagnosis—and introduces Gaussian

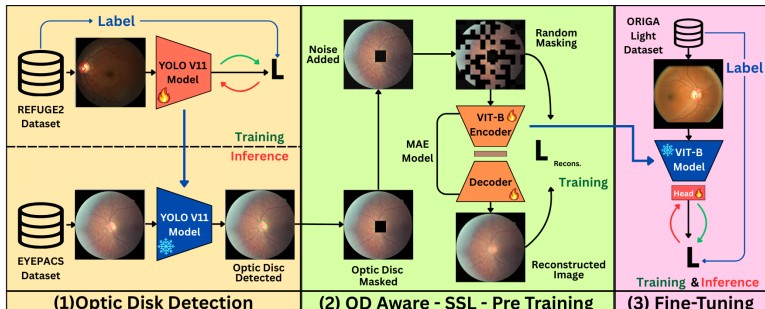

Figure 1: Overview of the proposed framework: (1) A YOLOv11 model is trained on REFUGE2 and used to detect optic discs (OD) in EYEPACS. (2) OD regions are masked and Gaussian noise is added for pretraining a Deconv-MAE with a ViT-B encoder. (3) The pretrained encoder is fine-tuned on ORIGA light for glaucoma classification.

noise to simulate real-world image degradations. By training a Deconvolutional Masked Autoencoder (Deconv-MAE) (Gao et al., 2022) to reconstruct clean images from these corrupted inputs, we encourage the model to learn robust, context-rich representations that are sensitive to the global retinal structure. The pretrained encoder is then fine-tuned for glaucoma classification on a limited annotated dataset, demonstrating superior performance over conventional supervised baselines and generic SSL models. This targeted design leverages anatomical priors to enhance learning efficiency and transferability in low-label regimes. Our approach is motivated by prior work (Huang et al., 2021; Kang et al., 2023), which highlights that due to the significant domain gap between natural and medical images, it is essential to leverage the distinctive characteristics of medical image modalities to design effective SSL methods.

## 2. Methodology

We propose a three-stage pipeline for OD-aware SSL, tailored for glaucoma classification. The framework emphasizes the optic disc —the anatomical region most indicative of early glaucomatous changes—and comprises: (1) OD Detection, (2) Pretraining using an OD-masked Deconv-MAE with noise-based corruption, and (3) Supervised Fine-tuning. Figure 1 illustrates the complete workflow.

**(1) Optic Disc Detection:** We train a YOLOv11 (Ultralytics, 2024) object detector on the REFUGE2 dataset (Fang et al., 2022), which includes bounding box annotations for the optic disc (OD). The model is trained for 100 epochs with a batch size of 16, image resolution of 640×640, a learning rate of 0.01, and a warmup of 3 epochs. Post-training, this detector is used on a filtered subset of the EYEPACS dataset (Gaid, 2023). Images with low-confidence or failed OD detection are discarded to ensure consistent masking quality.

**(2) OD-Aware SSL Pretraining:** We adopt a Deconv-MAE, based on Deblurring-MIM (Kang et al., 2023), using a ViT-B encoder. During pretraining, we apply two targeted perturbations: (i) the OD region is masked by zeroing its pixels, and (ii) Gaussian noise is added across the entire image to mimic real-world degradation. This setup encourages the encoder to rely on global retinal structure rather than localized high-saliency regions.

Table 1: Comparison of Supervised and SSL methods for glaucoma classification on ORIGA light. Performance is reported across architectures using Accuracy, Weighted F1 Score, Precision, and Recall. Precision and Recall are reported specifically for the suspicious class.

| Paradigm | Method | Architecture | Accuracy | Weighted F1 Score | Precision | Recall |
|---|---|---|---|---|---|---|
| | ResNet | ResNet 101 | 75.38 | 75.81 | 52.63 | 58.55 |
| Supervised | ViT | ViT-B | 70.77 | 70.47 | 43.75 | 41.18 |
| | ConvNext | ConvNext | 73.85 | 71.34 | 50.00 | 29.41 |
| | Swin Transformer | Swin L | 75.38 | 73.47 | 54.55 | 35.29 |
| SSL | MAE | ViT-B | 80.00 | 79.63 | 62.50 | **58.82** |
| | **(Ours)** | ViT-B | **84.62** | **83.42** | **81.81** | 52.94 |

Pretraining is conducted on 18,190 EYEPACS images (with 4,548 held out for testing), using a 40% random patch masking ratio in addition to OD masking. Training runs for 500 epochs with a batch size of 96, using the AdamW optimizer (lr=1.5e-4, weight decay=0.05, 40-epoch warmup). Similar setup is used for MAE.

**(3) Fine-tuning for Glaucoma Classification:** The pretrained encoder is fine-tuned on ORIGA-light (Zhang et al., 2010) for binary glaucoma classification using 520 training, 65 validation, and 65 test images. We use $224 \times 224$ resolution and the Adam optimizer (lr=1e−4) with batch sizes of 32 (supervised) and 64 (SSL). To reduce distributional shift—similar to DeblurringMIM—Gaussian noise is retained during SSL fine-tuning.

## 3. Experiments & Conclusion

We evaluate our approach using the ORIGA-light dataset (Zhang et al., 2010) for binary glaucoma classification. Following pretraining on the filtered EYEPACS subset (Gaid, 2023), we fine-tune the encoder on ORIGA-light and compare performance with both supervised models (ResNet101 (He et al., 2016), ViT-B (Dosovitskiy et al., 2021), ConvNext (Liu et al., 2022), Swin Transformer (Liu et al., 2021)) pretrained on ImageNet (Deng et al., 2009) and a standard ConvMAE baseline (Gao et al., 2022).

Our method outperforms all baselines across key metrics—achieving a 4.6% gain in accuracy and a 3.8% improvement in F1-score over vanilla MAE (see Table 1). This highlights the effectiveness of integrating lesion-aware masking and noise-based corruption during SSL pretraining. Notably, our approach achieves significantly higher precision, indicating improved confidence in positive glaucoma predictions, which is particularly valuable in clinical triage. Despite a slight drop in recall compared to standard MAE, the improved precision suggests that our model is more selective and robust in identifying glaucomatous cases. This trade-off reflects a more cautious decision boundary, potentially reducing false positives.

In conclusion, we present a novel OD-aware denoising masked autoencoder tailored for glaucoma classification from fundus images. By leveraging domain-specific priors—namely, masking the optic disc and simulating image degradation—our approach enables more effective self-supervised representation learning. Unlike conventional MAE protocols, we omit a separate linear probing stage to emphasize the value of end-to-end feature transfer in realistic downstream setups. Future work will explore extensions to multi-task settings such as OD/OC segmentation and validate generalization across broader datasets.

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
