# OpenReview forum: "Obscure to Observe: A Lesion-Aware MAE for Glaucoma Detection from Retinal Context"
_MIDL.io/2025/Short_Papers — MIDL 2025 - Short Papers_

### Official Review · Reviewer_Dtoi · 2025-04-26

**Rating:** 4
**Confidence:** 4

**Summary:**

This paper proposes a self-supervised learning (SSL) approach for glaucoma detection from retinal fundus images by injecting domain knowledge. The method is compared against supervised models (ResNet-101, ViT-B, ConvNeXt, Swin-Transformer) and a standard MAE. According to the authors, anatomically-informed masking and noise corruption enhance representation learning under low-label conditions.

The proposed “lesion-aware” MAE achieves improvements in accuracy and precision over baselines, indicating its potential value in medical imaging. However, the evaluation is somewhat limited in scope (single dataset, lack of ablations), and there are minor clarity issues. Overall, the novelty and promising results make this a solid contribution deserving acceptance, especially in a short-paper context.

**Strengths:**

1. Effectively incorporates domain-specific medical knowledge by explicitly masking the optic disc during pretraining.
2. Conducts sufficient experiments and comparisons, and according to the authors, achieves promising results.
3. The overall structure and narrative are clear.

**Weaknesses:**

1. Limited Evaluation and Missing Analysis: The method is evaluated only on a small single dataset (ORIGA-light) without external validation, lacks ablation studies to isolate the effects of OD masking and noise, does not report the accuracy of the YOLOv11-based OD detection, and omits discussion on computational costs. These limitations weaken the confidence in the method’s generalization and robustness.
2. Clarity and Performance Trade-offs: Minor writing issues (e.g., typos) slightly affect clarity; the method improves precision but sacrifices recall, raising concerns about missed glaucoma cases; and no comparisons are made with alternative SSL methods, which would strengthen the empirical validation.

---

### Decision · Program_Chairs · 2025-05-01

Accept